# Vernalization Procedure of Tuberous Roots Affects Growth, Photosynthesis and Metabolic Profile of *Ranunculus asiaticus* L.

**DOI:** 10.3390/plants12030425

**Published:** 2023-01-17

**Authors:** Giovanna Marta Fusco, Petronia Carillo, Rosalinda Nicastro, Giuseppe Carlo Modarelli, Carmen Arena, Stefania De Pascale, Roberta Paradiso

**Affiliations:** 1Department of Environmental, Biological and Pharmaceutical Sciences and Technologies, University of Campania Luigi Vanvitelli, 81100 Caserta, Italy; 2Department of Agricultural Sciences, University of Naples Federico II, 80055 Naples, Italy; 3Department of Biology, University of Naples Federico II, 80126 Naples, Italy

**Keywords:** geophytes, cold requirement, photosynthetic pigments, carbohydrates, amino acids, proteins, alanine

## Abstract

In *Ranunculus asiaticus* L., vernalization of propagation material is a common practice for the production scheduling of cut flowers, however little is known about the plant physiology and metabolism of this species as affected by cold treatments. We investigated the influence of two hybrids, MBO and MDR, and three preparation procedures of tuberous roots, only rehydration (control, C), and rehydration plus vernalization at 3.5 °C for 2 weeks (V2) and for 4 weeks (V4), on plant growth and flowering, leaf photosynthesis, and leaf metabolic profile in plants grown in pot in a cold greenhouse. Net photosynthesis (NP) was higher in MDR than in MBO. In the two genotypes, the NP did not change in V2 and increased in V4 compared to C in MBO, while was unaffected by vernalization in MDR. Quantum yield of PSII electron transport (ΦPSII), linear electron transport rate (ETR) and non-photochemical quenching (NPQ) did not differ in the two hybrids, whereas maximal PSII photochemical efficiency (Fv/Fm) was higher in MBO than in MDR. Fluorescence indexes were unaffected by the preparation procedure, except for ETR, which decreased in V2 compared to C and V4 in MDR. A significant interaction between genotype and preparation procedure was found in plant leaf area, which was reduced only in V4 in MBO, while decreased in both the vernalization procedures in MDR. In Control plants, flowering started in 65 days in MBO and 69 days in MDR. Compared to controls, both the vernalization treatments anticipated flowering in MDR, while they were detrimental or only slightly efficient in promoting flowering in MBO. Vernalization always reduced the quality of flower stems in both the hybrids.

## 1. Introduction

*Ranunculus asiaticus* L. (Family Ranunculaceae) is a perennial geophyte cultivated for cut flowers and potted plants production. Its biological cycle presents a vegetative development during the cool, moist winter season and a quiescence phase in the hot and dry summer months [1]. In the wild Mediterranean environment, the dry and dormant tuberous roots sprout in Autumn, when the first rain rehydrated the tissue, and develop a leaf rosette [2]. Flowering lasts from February to May, then the plant enters in dormancy, and the tuberous roots and the aerial part wilt and disappear in summer [3]. Contrary to many other geophytes, the shoot apical meristem in tuberous roots of *R. asiaticus* remains inactive during the quiescence, and it restarts when temperatures return mild, after rehydration. Kamenetsky et al. [4] showed that during root growth an increase in cortical cell size and an accumulation of pectin materials in the cell walls occur. In addition, the authors found that starch granules and protein bodies also accumulated to support the plant growth after summer quiescence. The tuberous roots are well adapted to afford long storage period, and these account for the identification of *R. asiaticus* as “resurrection geophyte” [4,5].

In Italy, *Ranunculus* is cultivated in cold greenhouse from rehydrated and vernalized tuberous roots, with planting from the end of August to the beginning of September and harvest from the end of November until the beginning of April. In cool, moist summer climate, flowering can be obtained in early November by sowing seeds in May, while in warmer climate the sowing in Spring is possible only when soil is artificially cooled at 18 °C [1]. However, tuberous roots allow an earlier flowering and provide a more abundant flower production compared to seedlings [3].

Tuberous roots prepared as propagation material are dehydrated to less than 15% moisture content. The genotype and the size of tuberous roots affect the flowering earliness and the number and features of flowers, and plants from bigger roots flower earlier and produce more and greater flowers compared to those from smaller roots [3].

Flowering in *Ranunculus* is a complex process influenced by the thermal history of tuberous roots and the photoperiod. Plants exhibit a low temperature requirement (night/day regime 5–10/12–25 °C, optimum day 16 °C) and a quantitative response to long day [2]. Cold treatments of tuberous roots (vernalization) anticipate sprouting, leaf rosette formation, and flowering and increase the number of flowers per plant compared to untreated roots [5]. This has been ascribed to the need for a cold period to break the summer vegetative dormancy, similarly to what occurs under natural growth conditions [4]. However, the response to vernalization can change among the hybrids. Low temperature induces starch breakdown, enhancing the content of free sugars like sucrose [6], and altering the levels of abscisic acid (ABA) and gibberellic acid (GA) [7]. This change in hormonal balance, consisting in a downregulation of ABA and an induction of the biosynthesis of GA active forms, determines dormancy breaking and promotes bulbs sprouting [6].

In *R. asiaticus* it is known an antagonistic activity between flowering and tuberization, and that these processes undergo a crosstalk with thermo- and photo- period. Short day (SD) and vernalization exert a positive control on meristematic activity enhancing the percentage of sprouting buds and increasing the number of leaves and flowers [3]. Whereas long day (LD) anticipates the flowering process inducing the growth of already formed buds, reducing the flowers yields and quality, and increasing the size of tuberous roots [3,8]. Temperature plays a very important role in these phenomena, also by interacting with photoperiod. In fact, under LD, the exposure to low temperature induces flowering, while higher temperatures result ineffective or detrimental [9,10].

Cultivation of *Ranunculus asiaticus* L. has been rising during the last years all over the world, also thanks to the breeding and the development of new hybrids, and vernalization of propagation material is a common practice for the production scheduling of cut flowers, as in many ornamental geophytes [1]. Despite this, little is known about the plant physiology of this species and, apart from some our recent studies [11,12,13,14], no information seems to be available on the primary and secondary metabolism as affected by cold treatments of propagation material. We investigated the influence of two hybrids of *R. asiaticus* L., characterized by different flowering earliness, and three preparation procedures of tuberous roots, including only rehydration and two times of exposure to cold temperature after rehydration, on photosynthesis, metabolic profile and flowering of plants grown in pot in cold glasshouse.

## 2. Results

### 2.1. Net Photosynthesis, Chlorophyll a Fluorescence and Leaf Photosynthetic Pigments

The light saturation curves evidenced that in both hybrids of *Ranunculus asiaticus* L., the rate of leaf net photosynthesis (NP) in Control, V2 and V4 plants increased with the level of the white light provided by halogen lamp, but the saturation for the two hybrids is reached at different PPFDs and appeared significantly affected by the preparation procedures. More specifically, the NP saturation in the hybrid MBO C and MBO V2 occurred at an irradiance of about 300 μmol photons m^−2^ s^−1^, with values of 3.56 and 3.23 μmol CO_2_ m^−2^ s^−1^, respectively, while in MBO V4 at 500 μmol photons m^−2^ s^−1^, reaching the values of 6.37 μmol CO_2_ m^−2^ s^−1^ (Figure 1). Moreover, in the whole range of tested irradiances, NP of MBO V4 has statistically higher (*p* ≤ 0.01) than MBO C and MBO V2, which showed comparable values. In the hybrid MDR, saturation of NP was obtained at the PPFD of 1000 μmol photons m^−2^ s^−1^, for MDR C and V2 and V4 with values of 8.06, 9.73 and 10.28 μmol CO_2_ m^−2^ s^−1^, respectively. Moreover, in the case of MDR hybrid, the V4 plants showed higher level (*p* ≤ 0.05) of NP than C and V2 ones in the whole range of PPFDs (Figure 1).

Measurements of NP at growth irradiance evidenced that only in rehydrated (Control), leaf net photosynthesis was higher in the hybrid MDR than in the hybrid MBO (Table 1). A significant interaction (*p* ≤ 0.05) was found between the genotype and the preparation procedure on net photosynthesis, which did not change in V2 and increased in V4 in MBO, while it was unaffected by vernalization in MDR, compared to Control (Table 1). Measurements of the chlorophyll *a* fluorescence emission evidenced no significant effect of the hybrid on quantum yield of PSII electron transport (Φ_PSII_), linear electron transport rate (ETR), and non-photochemical quenching (NPQ) (Table 1). Conversely, Fv/Fm ratio showed higher values (*p* ≤ 0.05) in MBO than in MDR (Table 1). The preparation procedure of tuberous roots did not affect the PSII photochemistry, however it influenced the ETR differently in the two hybrids. In fact, a significant interaction (*p* ≤ 0.05) was found between the genotype and the preparation procedure on ETR; while it did not change with vernalization in MBO, significantly decreased (*p* ≤ 0.05) in MDR subjected to V2 treatment compared to the other procedures (Table 1).

### 2.2. Plant Growth and Flowering

In Control plants, the number of leaves in fully developed plants was similar in the hybrids (23.2 leaves per plant on average), however the total leaf area was higher in MDR compared to MBO (+37%), even though this non determined significantly greater leaf dry matter and similar values were also recorded in root dry matter (Table 2).

Genotype and root preparation procedure significantly interacted on total plant leaf area, which was reduced significantly only under the V4 procedure in MBO plants (−82% compared to Control), while decreased after both the procedures in MDR (−34% in V2 and −72% in V4) (Table 2). Accordingly, the leaf dry weight strongly decreased only after the longer vernalization in MBO and after both the procedures in MDR (Table 2). The V4 also determined a significant reduction in dry matter accumulation in the roots (Table 2).

In plants from only rehydrated tuberous roots (C), the time for flowering seemed to be not different in the two hybrids, however data revealed a quite high variability in both the plant populations (92.5 ± 3.2 DAS in MDR C and 110.2 ± 5.3 in MBO C). Compared to only rehydration, the procedure V2 did not influence significantly flowering earliness in MBO (118.5 ± 4.0 DAS in MBO V2) and anticipated it in MDR (65.5 ± 5.6 DAS in MDR V2), while the procedure V4 anticipated flowering in both the hybrids (84.7 ± 11.2 DAS in MBO V4 and 34.3 ± 8.9 in MDRV4).

The quality of cut flowers did not differ between the hybrids while it was influenced by the preparation procedure of the propagation material. In control plants, cut flowers had similar stem length and diameter and consequently similar dry weight (Table 2). Compared to Control, vernalization of tuberous roots did not influence the stem height in MBO, while both the procedures reduced it in MDR (−17% and −33%, respectively) (Table 2). Despite the different genotype response to vernalization in terms of stem height, compared to control both the vernalization procedures determined in both the hybrids thinner stems and significantly lower flower stem fresh weight (data not shown) and dry weight (−42% and −66% in MBO, and −60% and −74% in MDR, in V2 and V4, respectively) (Table 2).

### 2.3. Leaves’ Metabolites Profiling

Glucose content in both hybrids MDR and MBO of *Ranunculus asiaticus* L. at flowering, differently from the vegetative phase, was significantly affected by *Hybrid*
**×**
*Vernalization procedure* (H **×** V). Specifically, in MDR the content of this sugar significantly increased after In V2 (+42%) and V4 (+37%) compared to control; while in V4 MBO glucose significantly decreased in V4 (−39%) compared to the Control (Table 3). Fructose content was significantly higher in in MBO than in MDR (on avg. 20.55 and 23.55 mg g^−1^ DW, respectively) at both vegetative phase and flowering (Table 3). In both growth phases, Vernalization Procedure (V) and H **×** V influenced sucrose content. In particular, in MDR, V2 and V4 showed an increase in sucrose content both at vegetative stage and flowering compared to respective Controls, being on average 0.26 and 0.12 µg mg^−1^ DW, respectively. In MBO, on the other hand, sucrose significantly decreased only in V2 treatment (−69% compared to Control). H **×** V was significant for starch content. In particular, starch increased in MDR V4 by 53% at vegetative stage and in MBO V2 by +54% at flowering, compared to respective Controls (Table 3).

The polyphenol content was affected by V and H **×** V, in both vegetative and flowering stages. At vegetative phase, its content decreased only in MBO V4 (−38%) compared to Control. Whereas at flowering, polyphenols increased in MDR V2 and decreased in MDR V4 (+16% and −22% compared to Control, respectively), while in MBO V4 these metabolites decreased by 36% compared to Control. Photosynthetic pigments, namely Chl *a*, Chl *b* and carotenoids, were significantly affected by Hybrid (H), V and H **×** V, both at the vegetative stage and flowering. Specifically, at the vegetative stage, Chl *a* significantly decreased in MBO V4 (−14%) compared to Control, while Chl *b* and carotenoids, whose contents were affected by H and H **×** V, significantly increased in MDR V2 and V4 by 30% and 25% on average, compared to respective Controls. At flowering, all three parameters were significantly affected by V, and only Chl *a* was also affected by H **×** V. In both hybrids, V4 treatment, showed significantly higher contents of Chl *b* and carotenoids (+20% and +14% in MDR, and 50% and +34% in MBO, compared to respective Controls). Chl *a* increased under V4 treatment (+33%) compared to Control only in the MBO hybrid (Table 3).

The soluble protein content at vegetative stage and at flowering, underwent variations depending on H and V. At vegetative stage, the protein contents in MBO and MDR 62.23 and 48.55 mg g^−1^ DW on average, respectively particularly MBO V4 showed the highest protein value, equal to 77.24 mg g^−1^ DW. At flowering, both in MDR and MBO the V4 treatment increased this parameter by 44% and 34% compared to respective Controls (Table 4). The variations in total amino acid profile in the two hybrids depended on H, V and the interaction between these two parameters. At the vegetative stage, MDR showed an amino acid content higher than that of MBO (on avg. 146.3 µmol g^−1^ and 97.00 µmol g^−1^, respectively), with V4 undergoing an increase in both MDR (+36%) and MBO (+69%) compared to respective Controls. At flowering, on the other hand, total amino acids decreased in MDR V4 (−41%), while increased in MBO V4 (+74%) compared with their Controls (Table 4).

Minor amino acids (including arginine, histidine, isoleucine, leucine, lysine, methionine, phenylalanine, tyrosine, tryptophan, and valine) were influenced only by the interaction H × V both in both the vegetative and flowering phases (Table 4 and Appendix A). Specifically, in the vegetative stage, Minor amino acids were significantly higher in MBO V2 than in the Control (14.71 µmol g^−1^ and 10.57 µmol g^−1^, respectively). At flowering, they were lower in MDR at V2 and V4 procedures (6.69 µmol g^−1^ and 5.40 µmol g^−1^) compared to Control (8.94 µmol g^−1^), while higher in MBO V4 compared to MBO C (12.64 µmol g^−1^ and 5.97 µmol g^−1^, respectively). In the vegetative phase, among minor amino acids (Appendix A) arginine increased in MDR V2 and MBO V2 (+67.9% and +63.8% respectively) as well as in MDR V4 (+80.4%) compared to respective Controls. In MBO V2 there was an increase in leucine, lysine, methionine and phenylalanine of 81.7%, 213%, 173%, 72%, compared to Controls, respectively; whereas tyrosine content increased from 0.62 µmol g^−1^ (control) to 1.04 µmol g^−1^(MBO V4). However, in MBO V4, there was also a decrease in isoleucine, leucine, methionine and phenylalanine of 28.4%, 57%, 80.8% and 58.8% compared to respective Controls (Appendix A). In MDR V4 the content of tryptophan raised of 123% compared to Control. In contrast, in the MDR V4 there was a decrease in isoleucine from 0.98 µmol g^−1^ to 0.76 µmol g^−1^, and also a decrease in phenylalanine in MDR V2 (−25%) and V4 (−36.7) compared with respective Controls (Appendix A).

In the flowering stage, in MDR V4 there was a decrease in arginine, isoleucine, leucine, phenylalanine, tryptophan and valine content of 38.2%, 37%, 37%, 71.4%, 74.3% and 58.7%, compared to respective Controls. Histidine and lysine decreased in MDR V2 (−40.4% and 27.4%, respectively) and V4 (−35.4% and 59.5%, respectively) compared to Controls. In contrast, in the hybrid MBO there was an increase in the content of minor amino acids, in both vernalization procedures. Specifically, arginine, isoleucine, leucine, lysine, methionine and valine content under V2 treatment raised of 41.4%, 54.1%, 27.5%, 71.4%, 55.6% and 53.5% compared to respective Controls. All the minor amino acids except tryptophan increased in MBO V4, in particular isoleucine, leucine, phenylalanine, tyrosine and valine raised of 445.8%, 192.5%, 417.3% and 250%, and 327.9% compared to respective Controls (Appendix A).

The H × V treatment affected branched chain amino acids (including isoleucine, leucine, valine; BCAAs) in the vegetative phase, with MBO showing an increase in V2 (+31%) and a decrease in V4 (−29%) compared to Control. In contrast, the content of BCAAs in MDR was unchanged (avg. 3.10 µmol g^−1^). At flowering, only MBO V4 tripled its content compared to Control, equal to 1.07 µmol g^−1^ (Table 4). H × V interaction influenced the content of various amino acids in both the vegetative and flowering state. In particular, alanine, arginine, asparagine, GABA, glutamate, glutamine, proline, serine, and tyrosine. For the MDR hybrid in the vegetative stage, an increase in alanine, arginine, GABA, glutamate, glutamine, and serine was observed under the V4 vernalization treatment compared with respective Controls (+97%, +80%, +54%, +78%, +19%, and +42%, respectively). The two amides glutamate, glutamine, and tyrosine increased in the MBO hybrid compared with Controls (+192%, +123% and +68%, respectively). Whereas alanine, arginine, GABA and serine increased under V2 vernalization treatment compared to Controls (+17%, +64%, +123% and +88%, respectively) (Table 4).

At flowering, alanine, glutamine, GABA and asparagine decreased in MDR V4 compared with controls (−43%, −23%, −65%, −92%, respectively). Moreover, in MDR V2 asparagine decreased (−67%) and GABA increased (+64%) compared to respective Controls. In MBO V2 e V4 GABA increased compared to control (+77% and 136%, respectively), on the other hand, an increase in the content of alanine, arginine, asparagine, glutamine, glycine, histidine, proline and tyrosine was observed compared with respective Controls (+83%, +124%, +222%, +50%, +121%, +30%, respectively) (Table 4).

## 3. Discussion

In our experiment, plants of two hybrids of *Ranunculus asiaticus* L. obtained by different vernalization procedures of tuberous roots were grown in the South of Italy from September till March, under short day condition in a cold greenhouse.

In the environmental and cultural conditions of our experiment, in Control plants from only rehydrated tuberous roots, flowering earliness did not change significantly between the hybrids, as observed in a previous experiment on plants of the same genotypes obtained with the same preparation procedure [13]. Cold treatments of rehydrated tuberous roots, similar to our V2 procedure, have been proved to anticipate flowering in different *Ranunculus* hybrids [5]; however, in our experiment the exposure to 3.5 °C for 2 weeks, which is the most common treatment in breeding farms, was effective in the hybrid MDR but ineffective or detrimental in MBO one. This confirms that the efficacy of vernalization depends on the hybrid-specific cold requirement [5]. Indeed, our results agree with the findings of Ohkawa [8] who studied the influence of different vernalization procedures (5 °C for 2 weeks and for 4 weeks compared to only rehydration) on flower initiation of 2 cultivars of *R. asiaticus*. Particularly, the author found a close relationship between flowering anticipation and the duration of cold treatment in one genotype, as we observed in MDR, while reported a negative influence of the 4 weeks thermal treatment on the flowering earliness/quality of flower stems [8].

In our experiment, flower stem characteristics were comparable in the two hybrids in plants obtained with only rehydration of tuberous roots, as expected on the basis of the technical sheets provided by the breeder. Vernalization reduced the fresh weight in both the hybrids and also the stem height in MDR, with stronger effect after 4 weeks of treatment, confirming negative effects on flower quality [8].

When net photosynthesis (NP) was measured at growth irradiance (343.3 ± 86.7 PPFD), neither difference between hybrids nor preparation procedure emerged, but the interaction was significant, in fact only in MBO the treatment V4 stimulated CO_2_ assimilation more than C and V2 did. The intrinsic differences in photosynthetic behaviour between MDR and MBO hybrids were evident by analysing the photosynthetic light response curves. MDR NP was higher than that of MBO in the whole range of tested irradiances, as already observed in previous experiments on the same genotypes in a growth chamber [11] and in a greenhouse [14]. However, the light curves indicate that the MBO hybrid was positively affected only by V4 treatment, conversely to MDR, where both V2 and V4 treatments stimulated photosynthesis compared to the control. The intrinsic differences between the hybrids became more evident as the irradiance increased. More specifically, under limiting light conditions, the carbon fixation process of both hybrids was not affected by preparation procedures, as confirmed by point measurements at not saturating growth irradiance. On the contrary, when NP approaches saturation (about 1000 PPFD), both preparation procedures V2 and V4 enhanced MDR NP compared to control, conversely only V4 sorted a stimulatory effect on MBO gas exchanges, suggesting that this hybrid needs prolonged vernalization for increasing NP. It is reasonable that the vernalization time may have differently altered the sink and source balance of carbohydrates and other metabolites in the two hybrids, as also demonstrated for other species [15].

Currently, measurements of photosynthesis of *Ranunculus asiaticus* L. are scarce in the literature, and the lack of information does not allow a direct comparison with our data. However, it is likely to suppose that the different behaviour of the two hybrids to vernalization procedures may depend on the intrinsic genetic basis [5] or regulatory mechanisms at the photosystem level determining a different light harvesting capability and conversion to PSII reaction centres [14]. Indeed, MDR compared to MBO was characterized by an elevated number of photosynthetic pigments, namely chlorophylls and carotenoids in vegetative phase that may have contributed to the better photosynthetic performance in this hybrid.

In terms of photochemistry, the elevated values of maximal PSII photochemical efficiency (Fv/Fm) in both MDR and MBO hybrids indicated a good photosystem healthy status [11], also confirmed by the quantum yield of PSII electron transport (Φ_PSII_) and non-photochemical quenching (NPQ) comparable between the hybrids. Therefore, the cold temperatures applied in this study in vernalization procedures did not affect the plant capability of light harvesting and utilization in photochemistry or thermal dissipation mechanisms. This result agrees with our previous research on the same hybrids treated with only two root preparation procedures (rehydration and rehydration plus vernalization for 2 weeks) [11] and indicates the absence of stress condition for photosynthetic apparatus in both control and plants from tuberous roots subjected to vernalization procedures [16,17,18]. However, it is noteworthy that, despite the good PSII functionality for both hybrids, the higher values of Fv/Fm ratio in MBO than MDR plants closer to the threshold of 0.80 may be an indicator of a higher potential capability by PSII of these plants to respond to putative stress.

MDR and MBO leaf metabolic profile under the three treatments showed a significant interaction between genotype and vernalization procedure in relation to the content of soluble sugars, chlorophyll, polyphenols, and amino acids (Figure 2). These metabolites, in particular sugars, are pivotal for plants because they can sustain growth and the development of flowers [19,20]. Besides, soluble sugars are not only important for their export to non-photosynthetic tissues and in metabolic events like oxidation to provide energy and carbon skeletons for biosynthetic reactions, but they can also function as signals to regulate gene expression [21]. Soluble sugars are also implicated in photosynthesis, osmolyte synthesis and sucrose metabolism [22]. The two vernalization procedures affected the content of soluble sugars, in particular increasing the content of sucrose in MDR, while only V2 decreased it in MBO in both the growth stages (Figure 2). Indeed, the increase in sucrose, induced in MDR at vegetative and flowering stage by both the vernalization procedures, could be involved in the promotion of flowering in *Ranunculus* as in several other species [23]. On the contrary, in MBO there was a stable higher average fructose content, while the transport sugar sucrose, particularly in MBO V2, decreased compared to MDR (Figure 2). In addition, MBO also accumulated polyphenols, secondary metabolites involved in antioxidant activity, which have very high cost in terms of energy consumption (50–70 moles ATP for mole) [24,25].Therefore, the increase in fructose and polyphenols in MBO during the vegetative stage was probably the symptom of an oxidative stress caused by vernalization and also the cause of the delay in flowering [26].

The presence of a higher stress in MBO V2 was also proved by the increase in BCAAs, that have an antioxidant activity and can function as alternative electron donor for the mitochondrial electron transport chain under oxidative stress [27]. This was not the only case in which the vernalization procedure affected the amino acids profile. In fact, also MDR V4 showed a higher content of alanine and glutamate demonstrating to have a better nitrogen use efficiency correlated to the higher growth capacity and flowering anticipation notwithstanding the longer cold treatment. Moreover, also the γ-aminobutyric acid (GABA) was present at higher concentrations compared to the control. This non protein amino acid, deriving from the decarboxylation of glutamate catalysed by glutamate decarboxylase, can exert a strong ROS scavenger activity thus protecting membranes and macromolecules from oxidative stress [28]. GABA and alanine are able to buffer the cytoplasmic acidosis and pH altered by the vernalization process as reported in [12], thus helping plants to cope with stress. In addition, the costs for their synthesis are compensated by the fact that after relief from oxidative stress alanine can be transaminated to pyruvate and converted to acetyl-Coenzyme A in the mitochondrial matrix, whereas GABA shunt can provide NADH and/or succinate to Krebs cycle playing an anaplerotic function [29] that can accelerate flowering in the subsequent growth stage. Finally, the longer vernalization procedure (V4) increased the protein content in both MDR and MBO hybrids. Accordingly, in a previous proteomic study of Zhou et al. [30], in which a comparison was done between plants of *Rhododendron* under normal temperature and cold stress, it was shown that under cold stress there was an accumulation of proteins, like DnaJ, HSP70, and Chaperonin 60, useful to protect and stabilize proteins and their folding, and protecting them from denaturation to avoid metabolism alteration under cold/oxidative stress.

## 4. Materials and Methods

The experiment was carried out at the Department of Agriculture of the University of Naples (Portici, Italy—40°49′ N, 14°20′ E), from 18 September 2018 to 30 March 2019.

Dry tuberous roots of two hybrids of *Ranunculus asiaticus* L., MBO (early flowering) and MDR (medium earliness) (Biancheri Creazioni, Italy, https://www.bianchericreazioni.it/ accessed on 1 October 2022) were subjected to three preparation procedures:-only rehydration: exposure to 12 °C for 24 h in humid chamber (Control, C);-rehydration followed by vernalization at 3.5 °C for 2 weeks (V2);-rehydration followed by vernalization at 3.5 °C for 4 weeks (V4).

Tuberous roots of the most common size for each hybrid were used (3–4 cm for MBO and 4–5 cm for MDR). Plants were grown in pot, on a mixture of perlite and peat (70:30 in vol.). Irrigation was alternated with fertigation (4 pulses per week in total). In the nutrient solution Hoagland full strength, pH and electrical conductivity (EC) were kept at 5.5 and 1.7 dS/m, respectively, and monitored with a portable pH-EC sensor (HI9813 series, Hanna Instruments Intl.). The mean values of air temperature and relative humidity (day/night) recorded during the experiment were 23.7 ± 5.0/12.30 ± 4.1 °C and 58.5 ± 6.8/74.3 ± 16.9%, respectively (Mean Value ± Standard Deviation).

### 4.1. Net Photosynthesis, Chlorophyll a Fluorescence and Leaf Photosynthetic Pigments

Net photosynthesis (NP), photochemical indexes and photosynthetic pigment content were determined in plants at vegetative stage, on mature leaves (8 weeks after planting), on 1 leaf per plant, in 4 plants per combination *Hybrid x Vernalization procedure*. Net photosynthesis (NP) was measured by a portable Infra- Red Gas Analyzer Walz HCM-1000 (Walz, Effeltrich, Germany) around midday (12:00−13:00). On leaves were carried out point measurements at growth irradiance as well as light response curves. In particular, the light saturation curves were performed in order to assess the capability of photosynthetic apparatus to reduce CO_2_ at sub saturating and saturating irradiance and to test if the potentiality of photosynthesis may be affected by vernalization procedure. For the measurements each leaf was darkened for 10 min and then exposed to increasing levels of light intensity (PPFD of 0, 50, 100, 250, 500, 1000, and 1500 μmol m^–2^ s^−1^). At each light intensity, the value of NP was taken at the steady state. The different PPFD on the leaf surface was obtained by a halogen lamp (1050-H, Walz) positioned on the cuvette plane of HCM-1000. The conditions inside the leaf chamber during the measurements were: temperature 20 °C, CO_2_ concentration 470 ppm, RH 67%, air flow rate 600 mL min^−1^. Photosynthesis was calculated by the software operating in HCM-1000 according to von Caemmerer and Farquhar [31].

Fluorescence measurements were conducted by a pulse amplitude modulated fluorometer (MINI-PAM, Walz, Effeltrich, Germany) on leaves previously analysed for gas exchanges. The maximal PSII photochemical efficiency (Fv/fm) was measured after darkening leaves for 45 min in the morning (9:00 to 10:00), with the following formula: Fv/Fm = (Fm − Fo)/Fm, in which Fv is the variable fluorescence and represents the difference between the maximal and the basal fluorescence (Fm − Fo). Measurements in the light were conducted around midday under environmental sunlight. The basal fluorescence (Fo) was induced in leaves by a weak light of about 0.5 µmol photons m^−2^ s^−1^, followed by a 1 s saturating light pulse of 8000 µmol photons m^−2^ s^−1^ to record the maximal fluorescence in the dark-adapted (Fm) and in the light-adapted (Fm’) state. The equation of Genty et al. [32] was used to calculate the quantum yield of PSII electron transport (Φ_PSII_), while the non-photochemical quenching (NPQ) was obtained according to Bilger and Björkman [33]. During gas exchange and fluorescence emission measurements in the light, the temperature in the glasshouse was 26.8 ± 2.8 °C, and the PPFD was on average 343.3 ± 86.7 µmol m^−2^ s^−1^ at canopy level (Mean value ± Standard deviation, n = 64).

Photosynthetic pigments were extracted from fully expanded leaves collected on plants, in the morning, at the rosette vegetative phase (11th week after planting), on 1 leaf per plants, in 3 plants per combination *Hybrid x Vernalization procedure*. Samples of 10 mg lyophilized leaf tissues were homogenized in 1 mL methanol, according to Annunziata et al. [34]. The resulting extracts were centrifuged at 4800 rpm for 15 min, and chlorophylls (Chl) *a* and *b* and total carotenoids were determined by measuring the absorbance of the supernatants at 470, 652 and 665 nm, in polypropylene microplates by a microplate reader (Synergy HT, BioTEK Instruments, Bad Friedrichshall, Germany), according to Woodrow et al. [27].

### 4.2. Plant Growth and Flowering

The number of leaves per plant was monitored weekly. Plant leaf area was estimated by non-destructive analysis of digital images of leaves with ImageJ software 1,50i version (Wayne Rasband National Institute of Health, USA), on 3 leaves per plant, on 5 plants per combination *Hybrid x Vernalization procedure*. The results obtained in fully grown plants at vegetative state (11 weeks after planting) are reported. The time for flowering was calculated as average of the number of days to obtain visible flower buds in 25 plants per treatment.

### 4.3. Metabolic Profile

Three fully expanded leaves per plant in 3 plants per combination *Hybrid x Preparation procedure* were sampled in the morning (9:00–11:00), during the 11th week after planting (vegetative phase) and the 16th week after planting (flowering phase). Leaves were immediately frozen in liquid nitrogen prior storage at −80 °C and, before the analysis, leaves samples were frozen dried at −50 °C for three days and powdered in a cooled mortar.

#### 4.3.1. Starch and Soluble Carbohydrate Analysis

Soluble sugars were extracted according to Dell’Aversana et al. [35] with some modifications. Lyophilized leaf material (10 mg) was extracted twice with 140 µL ethanol 80% (v:v) and once with 70 µL ethanol 50% (v:v) at 80 °C for 20 min. Tubes were cooled in ice and centrifuged at 14,000× *g* for 10 min at 4 °C. The clear supernatants of the three extractions were pooled together and stored at −20 °C until analysis. The pellets of the ethanolic extraction were heated at 90 °C for 2 h in 500 μL of 0.1 M KOH [36]. After cooling on ice, samples were acidified to pH 4.5, mixed 1:1 with a hydrolysis buffer containing sodium acetate 50 mM pH 4.8, α-amylase 2 U/mL and amyloglucosidase 20 U/mL, and incubated at 37 °C for 18 h. The samples were centrifuged at 14,000 rpm for 10 min at 4 °C, and the supernatant containing the glucose derived from starch hydrolysis was used for measurement. The content of glucose, fructose and sucrose in the ethanolic extracts, and the glucose derived by starch were determined by an enzymatic assay coupled with reduction in pyridine nucleotides and the increase in absorbance at 340 nm was recorded using a Synergy HT spectrophotometer (BioTEK Instruments, Bad Friedrichshall, Germany) according to Rouphael et al. [37]. The content of sugars was expressed as µmol g^−1^ DW.

#### 4.3.2. Soluble Proteins, Free Amino Acid Analysis

Proteins were extracted from an aliquot of lyophilized leaf samples (10 mg) with a buffer containing 200 mM TRIS-HCl pH 7.5 and 500 mM MgCl_2_ at 4 °C for 24 h. The clear supernatants (10 µL) were added to 190 µL of Protein Assay Dye Reagent Concentrate (Bio-Rad, Milan, Italy) diluted with ddH2O (1:5 v:v) [38]. The soluble protein content in the samples was calculated by comparison with standard curves obtained using known concentrations of bovine serum albumin (BSA) as the reference standard. Proteins were expressed as mg g^−1^ DW. Free amino acids (10 mg of lyophilized leaf samples) were extracted in 1 mL ethanol:water (40:60 v:v) overnight at 4 °C and determined by high-performance liquid chromatography (HPLC) after precolumn derivatization with ophthaldialdehyde (OPA) according to the method described by [37]. Proline was determined in the same ethanolic extract used for the amino acids determination by an acid ninhydrin colorimetric method according to Woodrow et al. [27].

#### 4.3.3. Polyphenols Analysis

The polyphenols content was evaluated according to Singleton et al. 1999 with some modifications. An aliquot of lyophilized leaf samples (30 mg) was extracted in 700 μL of 60% methanol (v:v), and 35 μL of extract were mixed with 125 μL of the Folin Ciocalteu reagent diluted with ddH2O (1:4 v:v) and, after mixing for 6 min, 650 μL of 3% (v:v) sodium carbonate was added. After 90 min at room temperature, the absorbance at 760 nm was determined in a microplate reader (Synergy HT, BioTEK Instruments). The polyphenols concentration was expressed as mg GAE g^−1^ DW (GAE are gallic acid equivalents) as described in Carillo et al. [39].

#### 4.3.4. Chlorophylls and Carotenoids Analysis

The analysis of photosynthetic pigments content (chlorophyll *a* and *b*, and carotenoids) was determined according to Wellburn [40]. An aliquot of 10 mg of lyophilized leaf samples was extracted in 1 mL methanol and centrifuged for 10 min at 13,500 rpm. The absorbances of the extracts at 665, 652, and 470 nm, was recorded using a Synergy HT spectrophotometer (BioTEK Instruments, Bad Friedrichshall, Germany) and used for the calculation of the contents of chlorophylls a and b and total carotenoids, expressed as µg g^−1^ DW.

## 5. Statistical Analysis

The experiment was conducted on 25 plants per combination *Hybrid x Vernalization procedures*. Data were subjected to statistical analysis by using SigmaPlot 12.0 (SPSS Inc. Norman Nie Dale Bent, Hadlai “Tex” Hull, Chicago, IL, USA) software package. The main effect of the categorical independent factors (i.e., hybrids and preparation procedures) and their interaction on the continuous dependent variables were analysed through the two-way ANOVA. In case of rejection of the null hypothesis, the Tukey’s HSD test was performed (*p* ≤ 0.05).

## 6. Conclusions

In conclusion, the two hybrids showed a different response to the root preparation procedure in terms of photosynthesis, plant growth and flowering earliness. These different responses can be attributed, as highlighted by the changes in the metabolic pathways, to the different effects exerted by the vernalization/cold procedures and their duration on the metabolism of the two hybrids. The MBO hybrid, more sensitive to cold treatment, decreased its NP efficiency and consequently the synthesis of sucrose, particularly under V2, probably for a lower ability to cope with photo-oxidative stress. Moreover, its need to continue synthetising polyphenols, comparable to Control, in addition to fructose and BCAAs diverts carbon skeletons and energy from the growth for a protective purpose. On the contrary, MDR both under V2 and V4 treatments can better cope with cold dependent oxidative stress, by synthetising sucrose, GABA and alanine. In fact, these compounds can act as compatible compounds and antioxidants, and at the same time be promptly re-used as source of carbon skeletons and energy as soon as the stress is relieved. These results highlight that the specific metabolic responses of the hybrids should be considered in the choice of the preparation procedure.

## Figures and Tables

**Figure 1 plants-12-00425-f001:**
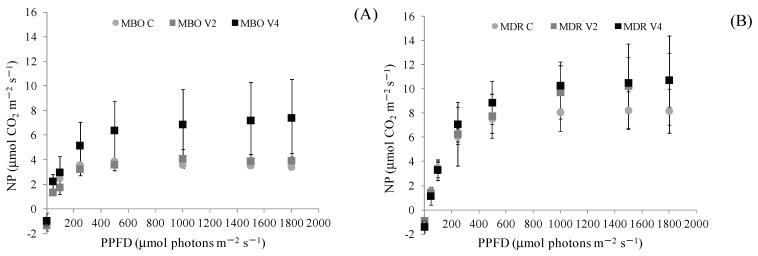
Light response curves of leaf net photosynthesis (NP) to increasing white light intensity in plants of *Ranunculus asiaticus* L. hybrids MBO (**A**) and MDR (**B**), obtained by three preparation procedures of tuberous roots, only rehydration (Control, C), rehydration plus vernalization for 2 weeks (V2), rehydration plus vernalization for 4 weeks (V4). Plants at vegetative phase (Week 8 from planting). Measurement conditions in the leaf chamber: 20 °C, RH 67%, 470 ppm CO_2_ concentration. Mean values ± Standard errors; n = 3.

**Figure 2 plants-12-00425-f002:**
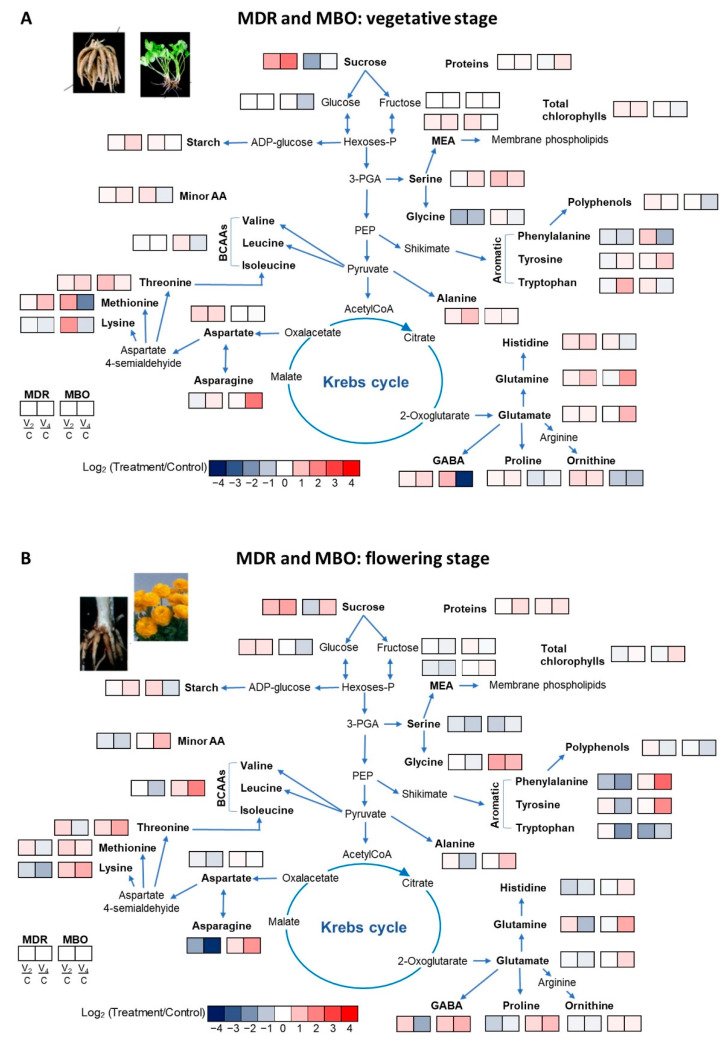
Pathway map summarizing the effect in vegetative (**A**) and flowering (**B**) stage in plants of *Ranunculus asiaticus* L. hybrids MDR and MBO, obtained by three preparation procedures of tuberous roots, only rehydration (Control, C), rehydration plus vernalization for 2 weeks (V2), rehydration plus vernalization for 4 weeks (V4). Plants at vegetative phase (Week 8 from planting). The heat map results were calculated as Logarithm base 1.5 (Log 1.5) V2/Control and V4/Control values for the two hybrids and visualized using a false-color scale, with red indicating an increase and blue a decrease. The significant ratios were indicated in the squares.

**Table 1 plants-12-00425-t001:** Net photosynthesis (NP) in µmol CO_2_ m^−2^ s^−1^, maximal PSII photochemical efficiency (Fv/Fm), quantum yield of PSII electron transport (Φ_PSII_), linear electron transport rate (ETR), non-photochemical quenching (NPQ), in plants of *Ranunculus asiaticus* L. hybrids MDR and MBO, obtained by three preparation procedures of tuberous roots, only rehydration (Control, C), rehydration plus vernalization for 2 weeks (V2), rehydration plus vernalization for 4 weeks (V4). Week 8 from planting (vegetative phase). Mean values ± Standard errors; n = 5; ns and * indicate non-significant or significant difference at *p* ≤ 0.05. Conditions in the glasshouse of measurements in the light: temperature 26.8 ± 2.8 °C, average PPFD at canopy level 343.3 ± 86.7 µmol m^−2^ s^−1^.

	MDR				MBO				Significance		
	C	V2	V4	Mean	C	V2	V4	Mean	H	V	H × V
NP	6.07 ± 0.66	4.35 ± 1.04	4.71 ± 1.12	5.05	3.57 ± 0.20	3.24 ± 0.53	7.42 ± 0.74	4.74	ns	ns	*
Fv/Fm	0.79 ± 0.01	0.79 ± 0.01	0.79 ± 0.00	0.79	0.79 ± 0.01	0.81 ± 0.01	0.80 ± 0.01	0.8	*	ns	ns
Φ_PSII_	0.42 ± 0.08	0.57 ± 0.03	0.51 ± 0.02	0.5	0.50 ± 0.04	0.50 ± 0.02	0.48 ± 0.03	0.49	ns	ns	ns
ETR	72.38 ± 6.45	54.12 ± 5.15	95.85 ± 9.06	74.11	86.65 ± 13.68	91.14 ± 4.89	85.69 ± 8.52	87.83	ns	ns	*
NPQ	1.59 ± 0.40	1.13 ± 0.18	1.29 ± 0.23	1.37	1.57 ± 0.19	1.52 ± 0.25	1.27 ± 0.10	1.46	ns	ns	ns

**Table 2 plants-12-00425-t002:** Plant growth parameters, flower stem characteristics and dry weight (D.W.) partitioning in *Ranunculus asiaticus* L. hybrids MBO and MDR obtained by three preparation procedures of tuberous roots, only rehydration (Control, C), rehydration plus vernalization for 2 weeks (V2), rehydration plus vernalization for 4 weeks (V4), and grown in cold glasshouse. Mean values ± Standard errors; n = 5; ns, * and ** indicate non-significant or significant difference at *p* ≤ 0.05 and *p* ≤ 0.01.

	MDR				MBO				Significance		
	C	V2	V4	Mean	C	V2	V4	Mean	H	V	H × V
Number of leaves (N./plant)	24.3 ± 3.3	27.7 ± 5.4	17.5 ± 0.3	23.2	24.0 ± 3.1	30.7 ± 6.7	15.0 ± 1.2	23.2	ns	*	ns
Total leaf area (cm^2^/plant)	883.6 ± 41.5	578.8 ± 33.2	242.8 ± 49.8	568.4	646.5 ± 100.8	458.8 ± 84.9	107.7 ± 4.5	406.9	**	**	*
Stem length (cm)	57.5 ± 2.7	47.9 ± 1.5	38.8 ± 2.3	48.1	59.4 ± 2.7	57.1 ± 2.7	59.3 ± 6.0	58.6	*	*	*
Stem diameter (mm)	5.25 ± 0.36	3.45 ± 0.16	3.85 ± 0.21	4.18	5.26 ± 0.23	4.04 ± 0.30	2.64 ± 0.49	3.98	ns	*	ns
Leaf D.W. (g/plant)	3.32 ± 0.23	2.74 ± 0.47	0.92 ± 0.17	2.36	3.10 ± 0.45	2.18 ± 0.37	0.30 ± 0.04	1.86	ns	*	ns
Root D.W. (g/plant)	0.61 ± 0.12	0.32 ± 0.07	0.12 ± 0.03	0.36	0.48 ± 0.14	0.27 ± 0.10	0.08 ± 0.01	0.28	ns	*	ns
Flower stem D.W. (g/plant)	4.26 ± 0.54	1.70 ± 0.14	1.12 ± 0.12	2.36	3.25 ± 0.29	1.87 ± 0.17	1.10 ± 0.47	2.07	ns	*	ns

**Table 3 plants-12-00425-t003:** Glucose, fructose, sucrose and starch (in mg g^−1^ DW), chlorophyll a (Chl *a*), chlorophyll b (Chl *b*), carotenoids, and polyphenols (in µg g^−1^ DW) in plants of *Ranunculus asiaticus* L. hybrids MDR and MBO, obtained by three vernalization procedures of tuberous roots, only rehydration (Control, C), rehydration plus vernalization for 2 weeks (V2), rehydration plus vernalization for 4 weeks (V4). Week 8 from planting (vegetative phase). ns, *, ** and ***; indicate non-significant or significant difference at *p* ≤ 0.05, *p* ≤ 0.01, *p* ≤ 0.001, respectively. Different lowercase or capital letters within each row, for specific vernalization procedure, indicate significant differences at *p* ≤ 0.05.

	MDR				MBO				Significance		
	C	V2	V4	Mean	C	V2	V4	Mean	H	V	H × V
*Vegetative phase*											
Glucose	35.63	35.17	35.94	35.58	37.32	39.02	19.66	32.00	ns	ns	ns
Fructose	20.75	20.86	21.08	20.89 A	22.82	23.59	23.47	23.29 B	**	ns	ns
Sucrose	0.05 a	0.14 b	0.24 c	0.14	0.16 b	0.05 a	0.14 b	0.12	ns	*	*
Starch	19.22 a	21.17 a	29.39 b	23.26	24.64 c	28.11 bc	25.29 c	26.01	ns	ns	***
Polyphenols	7.81 a	9.39 a	8.37 a	8.53	11.34 b	11.93 b	7.06 a	10.12	ns	*	***
Chl *a*	0.18 a	0.21 a	0.23 a	0.21 A	0.17 a	0.18 a	0.15 b	0.17 B	***	ns	*
Chl *b*	0.05 a	0.07 b	0.06 b	0.06 A	0.05 a	0.05 a	0.04 a	0.05 B	**	ns	***
Carotenoids	0.04 a	0.05 b	0.05 b	0.05 A	0.04 a	0.04 a	0.04 a	0.04 B	***	*	***
*Flowering phase*											
Glucose	19.9 a	28.27 b	27.32 b	25.16	35.7 b	35.18 b	21.71 a	30.87	ns	ns	*
Fructose	21.50	20.95	18.07	20.17 A	22.72	25.55	21.45	23.24 B	*	ns	ns
Sucrose	0.05 a	0.10 b	0.14 b	0.10	0.09 b	0.05 a	0.16 b	0.10	ns	***	*
Starch	19.82 a	21.02 a	28.2 a	23.01	25.56 a	39.31 b	17.55 a	27.47	ns	ns	**
Polyphenols	12.87 a	14.97 b	10.05 c	12.63	13.76 ab	12.68 a	8.82 c	11.75	ns	**	***
Chl *a*	0.17 a	0.15 b	0.19 a	0.17	0.15 b	0.15 b	0.20 a	0.17	ns	**	*
Chl *b*	0.05 a	0.04 a	0.06 b	0.05	0.04 a	0.04 a	0.06 b	0.05	ns	**	ns
Carotenoids	0.04 a	0.03 a	0.04 b	0.04	0.03 a	0.03 a	0.04 b	0.04	ns	*	ns

**Table 4 plants-12-00425-t004:** Soluble proteins (in mg g^−1^ DW) and free amino acids (in µmol g^−1^ DW) in plants of *Ranunculus asiaticus* L. hybrids MDR and MBO, obtained by three vernalization procedures of tuberous roots, only rehydration (Control, C), rehydration plus vernalization for 2 weeks (V2), rehydration plus vernalization for 4 weeks (V4). Week 8 from planting (vegetative phase). Amino acids (AA), γ-aminobutyric acid (GABA), monoethanolamine (MEA), branched-chain amino acids (BCAAs). ns, *, ** and ***; indicate non-significant or significant difference at *p* ≤ 0.05, *p* ≤ 0.01, *p* ≤ 0.001, respectively. Different lowercase or capital letters within each row, for specific vernalization procedure, indicate significant differences (*p* ≤ 0.05).

		MDR			MBO				Significance		
	C	V2	V4	Mean	C	V2	V4	Mean	H	V	H × V
*Vegetative phase*											
Soluble proteins	46.26 a	48.50 a	50.89 a	48.55 A	57.58 a	51.86 a	77.24 b	62.23 B	*	ns	*
Alanine	2.75 a	3.41 a	5.43 b	3.86	3.52 a	4.11 b	3.96 a	3.86	ns	*	*
Asparagine	49.03 a	40.67 a	62.11 a	50.61 A	6.32 b	6.84 b	28.29 b	13.82 B	***	ns	*
Aspartate	7.82	11.85	11.67	10.45	10.69	10.74	10.27	10.56	ns	ns	ns
GABA	7.37 a	8.72 ab	11.36 bc	9.15	6.10 a	13.59 c	0.47 d	6.72	ns	ns	***
Glutamate	14.98 a	17.39 a	26.71 c	19.69	7.95 b	7.33 b	23.22 c	12.83	ns	**	***
Glutamine	25.46 a	27.84 a	30.4 b	27.90	16.84 c	17.16 c	37.50 b	23.83	ns	***	***
Glycine	1.45 a	0.61 b	0.67 b	0.91 A	0.54 b	0.62 b	0.46 b	0.54 B	*	ns	**
MEA	3.70	4.48	4.91	4.36	3.98	5.49	4.03	4.50	ns	ns	ns
Ornithine	0.46 a	0.69 b	0.63 b	0.60 A	0.63 b	0.32 ac	0.30 c	0.42 B	*	ns	***
Proline	1.96	2.21	2.35	2.18	2.18	1.55	1.83	1.85	ns	ns	ns
Serine	4.12 a	3.92 a	5.87 b	4.64	4.12 a	7.74 c	6.06 b	5.98	ns	*	***
Threonine	0.69 a	0.84 b	1.06 c	0.86	0.64 ab	1.24 c	0.78 a	0.89	ns	*	***
Total AA	129.6 a	133.4 a	175.8 b	146.3 A	74.08 c	91.45 c	125.5 a	97 B	***	*	***
Minor AA	9.76 ad	10.80 ac	12.60 ab	11.05	10.57 ad	14.71 b	8.30 cd	11.19	ns	ns	**
BCAAs	3.14 ab	3.02 ab	3.13 ab	3.10	3.20 b	4.20 c	2.28 a	3.23	ns	ns	*
*Flowering phase*											
Soluble Proteins	31.58 a	32.71 a	45.43 b	36.57	33.51 a	40.39 a	45.06 b	39.65	ns	*	ns
Alanine	2.22 ab	2.47 a	1.27 b	1.99	1.97 ab	2.05 ab	3.61 c	2.54	ns	ns	**
Asparagine	7.02 a	2.34 b	0.56 c	3.31	4.04 b	5.72 a	13.03 d	7.60	*	ns	**
Aspartate	6.41 ac	5.34 b	4.29 c	5.35 A	7.99 a	8.64 a	7.77 ab	8.14 B	***	ns	*
GABA	1.12 a	1.84 b	0.39 c	1.11	1.08 a	1.91 b	2.55 d	1.85	ns	ns	***
Glutamate	10.98	15.53	4.81	10.44	9.74	8.98	25.04	14.59	ns	ns	ns
Glutamine	21.34 ab	19.86 a	16.49 c	19.23	16.28 ac	15.97 bc	24.44 d	18.89	ns	ns	***
Glycine	0.21 ab	0.21 b	0.17 b	0.20	0.14 b	0.38 c	0.31 ac	0.28	ns	ns	**
MEA	3.92	3.06	2.73	3.24	3.11	3.20	3.49	3.27	ns	ns	ns
Ornithine	0.57	0.50	0.49	0.52	0.54	0.64	0.64	0.60	ns	ns	ns
Proline	2.31 ab	1.30 d	1.90 ac	1.83	1.17 d	1.79 bcd	2.38 a	1.78	ns	ns	***
Serine	3.77 ac	2.73 bc	2.10 c	2.87 A	5.96 a	3.31 ac	4.76 ab	4.67 B	**	ns	*
Threonine	0.47 a	0.70 b	0.35 a	0.51	0.30 a	0.43 a	0.77 b	0.50	ns	ns	**
Total AA	69.28 a	62.59 a	40.96 b	57.61	58.27 ab	59.38 ab	101.44 c	73.03	ns	ns	***
Minor AA	8.94 a	6.69 b	5.40 b	7.01	5.97 b	6.36 b	12.64 c	8.32	ns	ns	***
BCAAs	1.65 ab	1.64 ab	0.85 a	1.38	1.07 a	1.54 ab	4.31 b	2.31	ns	ns	*

## Data Availability

The data are contained within the article and in the Appendix A.

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
