# Peer review of "Vernalization Procedure of Tuberous Roots Affects Growth, Photosynthesis and Metabolic Profile of Ranunculus asiaticus L."

_plants, 2023, doi:10.3390/plants12030425_

Round 1

Reviewer 1 Report

Dear Editor,

Please receive my review for the manuscript entitled “Vernalization procedure of tuberous roots affects growth, photosynthesis and metabolic profile of Ranunculus asiaticus L.  by Fusco et al. (2022).

The authors studied the effect of tuberous roots preparation on growth, flowering, photosynthesis, and metabolic profile. The authors found that the two hybrids responded differently to the treatments, and the treatment and hybrid interacted between each other for the studied parameters. The manuscript is well written, contains useful information on this subject for the scientific communities and growers/horticulturists in general. Although I recommend accepting the manuscript, the below points should be addressed before manuscript acceptance.

Abstract:

-The authors should add at the beginning of the abstract why this research was conducted (the rational of the research).

- The authors mentioned  that “A significant interaction between genotype and preparation procedure was found in plant leaf area, which was reduced only in V4 in MBO, while decreased in both the vernalization procedures in MDR”. To show a significant interaction between  genotype and preparation procedure, an ANOVA table should be presented and included in the manuscript.

-Some scientific names should be italic: Please revise throughout the manuscript and correct as appropriate.

Results:

-Figure 1B: The Y-axis label is not clear; please correct.

-Figure 2 does not exist. There is Figure 1 and Figure 3; where is figure 2?

- All abbreviations in Tables titles or Figures titles should be defined and explained as either a footnote or defined in the title.

Reference list:

-Some journals were abbreviated some not; please stick to the instructions to the journal instructions.

Author Response

R1: Reviewer 1

OR: our response

Abstract:

R1 - The authors should add at the beginning of the abstract why this research was conducted (the rational of the research).

OR – We thank the reviewer for this suggestion. The objective of the experiment has been added.

R1 - The authors mentioned that “A significant interaction between genotype and preparation procedure was found in plant leaf area, which was reduced only in V4 in MBO, while decreased in both the vernalization procedures in MDR”. To show a significant interaction between genotype and preparation procedure, an ANOVA table should be presented and included in the manuscript.

OR – We thank the reviewer for highlighting this oversight. The interaction between genotype and preparation procedure in plant leaf area has been highlighted in Table 2.

R1 - Some scientific names should be italic: Please revise throughout the manuscript and correct as appropriate.

OR –The scientific names have been corrected.

Results:

R1 - Figure 1B: The Y-axis label is not clear; please correct.

OR – Done.

R1 - Figure 2 does not exist. There is Figure 1 and Figure 3; where is figure 2?

OR – The numbers of figures have been corrected.

R1 - All abbreviations in Tables titles or Figures titles should be defined and explained as either a footnote or defined in the title.

OR – The abbreviations have been included in the caption of table 4.

Reference list:

R1 - Some journals were abbreviated some not; please stick to the instructions to the journal instructions.

OR – The names of journals have been written correctly according to the journal instructions.

Reviewer 2 Report

The study focusing on the effect of vernalization on growth of Ranunculus asiaticus is seemly interesting. Substantial work was performed by the authors. However, there are still some important questions that could impede the publication of the manuscript. 

(1). the authors wanted to investigate the influence of dehydration and vernalization on tuberous root growth, but they ignored the process of sprouting, and just measure the indexs of photosynthesis 8 weeks after planting.  It is difficult to evaluate whether vernalization affects photosynthesis and related metabolism in mature plants. This is maybe why the authors did not find significant difference in many aspects.

(2). The introduction is wordy, trying to emphasize key points would be better.

(3) in the tables, there is no unit for each index.

(4) For the Figure 3, too much metabolic informaiton is lacking, the authors should revise it to summarize important results without any exaggeration. 

Author Response

R2: Reviewer 2

OR: our response

R2 - The study focusing on the effect of vernalization on growth of Ranunculus asiaticus is seemly interesting. Substantial work was performed by the authors. However, there are still some important questions that could impede the publication of the manuscript.

R2 - (1). the authors wanted to investigate the influence of dehydration and vernalization on tuberous root growth, but they ignored the process of sprouting, and just measure the indexs of photosynthesis 8 weeks after planting. It is difficult to evaluate whether vernalization affects photosynthesis and related metabolism in mature plants. This is maybe why the authors did not find significant difference in many aspects.

OR - We thank the reviewer for this observation which allows us to clarify this critical issue. It is noteworthy that plants are well developed when the measurements of photosynthetic indexes were performed, but they are still in the vegetative phase. Generally, the photosynthetic parameters (gas exchanges and photochemistry) are measured not during sprouting but in the vegetative stage. This is because, at this stage, the photosynthetic process reaches maximum performance. If some modifications occurred following dehydration and vernalization (i.e., impairment of PSII electron transport or stomatal limitation), they would have been detectable also at the time of eco-physiological measurements trigging changes in the secondary carbon metabolism.

R2 - (2). The introduction is wordy, trying to emphasize key points would be better.

OR – Done.

R2 - (3) in the tables, there is no unit for each index.

OR - We thank the reviewer for highlighting this oversight. The units of measure have been added in Table 2.

R2 - (4) For the Figure 3, too much metabolic information is lacking, the authors should revise it to summarize important results without any exaggeration.

OR - In Figure 2 (previously Fig. 3) the metabolites (quantitatively measured) starting from their precursors (also quantitatively determined) are shown. It follows the same graphical concept of MAPMAN (Thimm O, Bläsing O, Gibon Y, Nagel A, Meyer S, Krüger P, Selbig J, Müller LA, Rhee SY, Stitt M. MAPMAN: a user-driven tool to display genomics data sets onto diagrams of metabolic pathways and other biological processes. Plant J. 2004 Mar;37(6):914-39. doi: 10.1111/j.1365-313x.2004.02016.x. PMID: 14996223). Moreover, it is not the first time we use this kind of representation as possible to see in the following manuscripts:

- Carillo P, Dell’Aversana E, Modarelli GC, Fusco GM, De Pascale S and Paradiso R (2020) Metabolic Profile and Performance Responses of Ranunculus asiaticus L. Hybrids as Affected by Light Quality of Photoperiodic Lighting. Front. Plant Sci. 11:597823.

-Carillo, P.; Kyratzis, A.; Kyriacou, M.C.; Dell’Aversana, E.; Fusco, G.M.; Corrado, G.; Rouphael, Y. Biostimulatory Action of Arbuscular Mycorrhizal Fungi Enhances Productivity, Functional and Sensory Quality in ‘Piennolo del Vesuvio’ Cherry Tomato Landraces. Agronomy 2020, 10, 911.

Reviewer 3 Report

This work well presented the interaction of hydration and vernalization in Ranunculus asiaticus L., and elucidated the mechanisms from metabolite perspective.

The word "anticipate" flowering in the text was confusing. I was not sure that authors mean "forward" flowering?

Author Response

R3: Reviewer 3

OR: our response

R3 - The word "anticipate" flowering in the text was confusing. I was not sure that authors mean "forward" flowering?

OR - Scheduling of plant production is a critical aspect in the modern floriculture, since nowadays sales of cut flowers and ornamentals are not oriented to the recurring holidays as in the past, but always more to impulse buying, implying a more diverse and constant demand on the market. This requires a continuous production, hence diverse techniques to modulate the duration of the growing cycle, by hastening or slowing down the plant growth and development, have been developed to match the plant flowering to the market demand. Among the numerous approaches, manipulation of climatic parameters (e.g., temperature before or after planting) in the growth environment is one of the most common in greenhouse floriculture and, in general, the control of the storing temperature of bulbs is the most effective in geophytes with only one flowering.

On this basis, the efficiency of the strategies to promote flowering is often expressed in terms of their ability to promote flowering and “to anticipate” is a common term to describe this effect (see Proietti S., Scariot V., De Pascale S., Paradiso R., 2022. Flowering mechanisms and environmental stimuli for flower transition: bases for strategies of production scheduling in greenhouse floriculture. Review article. Plants, 11, 432. https://doi.org/10.3390/plants11030432).
